# Application of Chitosan-Based Molecularly Imprinted Polymer in Development of Electrochemical Sensor for *p*-Aminophenol Determination

**DOI:** 10.3390/polym15081818

**Published:** 2023-04-07

**Authors:** Ani Mulyasuryani, Yuniar Ponco Prananto, Qonitah Fardiyah, Hanandayu Widwiastuti, Darjito Darjito

**Affiliations:** 1Chemistry Department, Faculty of Mathematics and Natural Sciences, Universitas Brawijaya, Jl. Veteran, Malang 65145, Indonesia; 2Pharmaceutical and Food Analysis Department, Health Polytechnic, Jl. Besar Ijen 77C, Malang 65112, Indonesia

**Keywords:** molecularly imprinted polymer, screen-printed carbon electrode, cyclic voltammetry, square wave voltammetry, *p*-aminophenol, chitosan, glutaraldehyde, sodium tripolyphosphate

## Abstract

Molecularly Imprinted Polymers (MIPs) have specific recognition capabilities and have been widely used for electrochemical sensors with high selectivity. In this study, an electrochemical sensor was developed for the determination of *p*-aminophenol (*p*-AP) by modifying the screen-printed carbon electrode (SPCE) with chitosan-based MIP. The MIP was made from *p*-AP as a template, chitosan (CH) as a base polymer, and glutaraldehyde and sodium tripolyphosphate as the crosslinkers. MIP characterization was conducted based on membrane surface morphology, FT-IR spectrum, and electrochemical properties of the modified SPCE. The results showed that the MIP was able to selectively accumulate analytes on the electrode surface, in which MIP with glutaraldehyde as a crosslinker was able to increase the signal. Under optimum conditions, the anodic peak current from the sensor increased linearly in the range of 0.5–35 µM *p*-AP concentration, with sensitivity of (3.6 ± 0.1) µA/µM, detection limit (S/N = 3) of (2.1 ± 0.1) µM, and quantification limit of (7.5 ± 0.1) µM. In addition, the developed sensor exhibited high selectivity with an accuracy of (94.11 ± 0.01)%.

## 1. Introduction

Molecular Imprinting Technology (MIT) is currently a synthetic approach to designing molecular recognition materials with the capability of mimicking natural recognition as biological receptors. MIT applications include separation and purification, chemical sensors, catalysis, and receptor systems [1,2,3,4,5,6,7]. MIT is based on the formation of complexes between analytes (templates) and functional monomers. In the presence of an excess crosslinking agent, a three-dimensional polymer network is formed. After the polymerization process, the template is removed from the polymer, leaving specific recognition sites that complement each other in shape, size, and chemical function to the template molecule [8,9,10,11]. The polymerization process is not easy to carry out; thus, several MIPs have been developed from functional polymers by adding controlled crosslinkers. Functional polymers that have been developed as MIP membranes and applied to electrochemical sensors are chitosan, which was used for MSG (monosodium glutamate) sensors [12], poly vinyl alcohol (PVA) for paracetamol sensors [13], and arrowroot starch-PVA for acid sensors [14]. In this study, chitosan–PVA was used as a functional polymer, glutaraldehyde and sodium tripolyphosphate (STPP) as the crosslinkers, and acetic acid as a catalyst.

*Para*-Aminophenol (*p*-AP) is the main degradation product of paracetamol and the main impurity of its synthesis process, which, by law, should not exceed the threshold of 0.005% and 0.1% by weight [15]. Acetaminophen or paracetamol (PA) is an analgesic, commonly available as a single drug or in combination preparations, which are formulated in various dosage forms, including tablets, syrups, and soluble powders. Paracetamol has substantial antipyretic activity and is often included in over-the-counter and prescription drug therapies for common ailments, such as headaches and fever [16]. These drugs are widely available not only in pharmacies but also in patent drugstores and supermarkets, in which their handling and storage conditions can be far from ideal. Improper storage of the drug can result in *p*-AP [17] at levels that exceed the threshold. Moreover, the presence of *p*-AP in paracetamol samples may not only be a cause of therapeutic failure, but also may pose a safety issue, since *p*-AP has been shown to be significantly more potent than paracetamol as a nephrotoxicant in animal models [18]. The general method for determining *p*-AP levels is by spectrophotometry [19,20], and a Flow Injection Analysis (FIA) technique with a developed UV detector [21]. Since *p*-AP is an electroactive compound, its determination will be more selective by electrochemical methods. Both *p*-AP and PA can be oxidized at a potential of 0–0.6 volts [22]; therefore, to increase the selectivity and sensitivity of *p*-AP detection, MIP membranes are used as modifiers on carbon electrodes. In this respect, electrochemical methods offer the benefits of elevated sensitivity, accuracy, convenient operation, cheap instrumentation, facile integration, and portability [23,24].

The *p*-AP electrochemical sensor has been widely developed by differential pulse voltammetry (DPV) with various carbon-modified working electrodes with nanomaterials [22,25,26,27], but generally, it is used for simultaneous determination of PA. Individually, the determination of *p*-AP has been developed electrochemically based on the modification of carbon electrodes by nanoparticles [28,29,30]. The individual determination of *p*-AP was carried out using glassy carbon modified by Fe_3_O_4_-Au/MOF nanocomposite. The electrode exhibits optimal catalysis at a temperature of 50 °C and scanning rate of 0.1 V s^−1^ in pH buffer 5. The equation for the calibration plot was |I_pa_| = 2.7372C + 2.1068, and R^2^ was 0.9952, with the detection limit of 0.38 mol L^−1^ (S/N = 3) in *p*-Ap concentration range of 0.1–10 mmol L^−1^ [31]. The detection system has a very good recovery rate and anti-interference, but in this system, it is necessary to regulate the operating temperature and an oxidase enzyme must be added. It becomes less practical to develop into a *p*-AP sensor for field analysis. In this research, a *p*-AP sensor was developed from a modified screen-printed carbon electrode (SPCE) with a chitosan-based MIP.

Several studies used chitosan for MIP preparation [32,33,34]. Chitosan is a natural biopolymer that has non-toxic properties and is available sustainably. Chitosan has two active groups in its structure, namely an amine (-NH_2_) group and a hydroxyl (-OH) group [35]. The amine group in chitosan can be modified into a secondary amine group (-NHR) by binding it to other compounds or groups [35,36]. Chitosan can undergo cross-linking reaction with sodium tripolyphosphate (STPP) [37,38] and glutaraldehyde. Both glutaraldehyde and STPP can function as chitosan crosslinkers in acidic conditions in acetic acid [35]. Furthermore, chitosan has the ability to form stable films on solid substrates [39]. In this research, a chitosan-based MIP was developed, in which *p*-AP was used as the template, with STPP and glutaraldehyde as the crosslinkers. Identification of MIP molecules was conducted by FTIR, whereas surface morphology of MIP was examined by SEM.

SPCE is an electrochemical measurement device which is produced by printing various types of inks on a plastic or ceramic substrate. The three-electrode SPCE system consists of a carbon working electrode (WE), a silver pseudo-reference electrode (RE), and a carbon counter-electrode (CE). SPCE offers several advantages, such as extensive work surface modification capacity, and possible connection to portable instrumentation for in situ determination of specific analytes. In addition, compared to the traditional three-electrode system, the volume of analyte for SPCE is much less than that of the traditional systems for electrochemical analysis. Several SPCE applications have been developed, including in the fields of biosensors, environmental protection, and drug and food safety [40]. In this study, the SPCE was used because it can be developed as a disposable sensor.

In other words, this research aimed to apply the natural polymer-based MIP technology for the development of electrochemical sensors. In this study, the chitosan was the basic polymer, *p*-AP was the template, and glutaraldehyde and STPP were the crosslinkers. The electrochemical sensor developed here was SPCE modified with MIP, which was then used for the determination of *p*-AP in paracetamol samples. This sensor is expected to be developed for a disposable sensor.

## 2. Method and Materials

### 2.1. Chemical Reagents and Apparatus

The reagents used were of high purity and used as received. Double-distilled water was used for all tests. Chemicals used include: *p*-AP (CAS 123-30-8, Darmstadt, Germany), chitosan (65% deacetylation, Bandung, Indonesia), polyvinyl alcohol (CAS 9002-89-5, Darmstadt, Germany), 50% glutaraldehyde (CAS 111-308, Darmstadt, Germany), sodium tripolyphosphate (CAS 6132-04-3, Darmstadt, Germany), citric acid (CAS 5949-29-1, Darmstadt, Germany), phosphoric acid (CAS 7664-38-2, Darmstadt, Germany), disodium hydrogen phosphate (CAS 7762-85-6, Darmstadt, Germany), trisodium phosphate (CAS 10101-89-0, Darmstadt, Germany), acetic acid (CAS 64-19-7, Darmstadt, Germany), HCl (CAS 7647-01-0, Darmstadt, Germany), NaOH (CAS 1310-73-2, Darmstadt, Germany). The equipment used in this study was general laboratory glassware, µStat200 drops (Metrohm, Oviedo, Spain), SPCE DRP110, 4 mm diameter of WE (DropSens, Oviedo, Spain), IRSpirit-T (Shimadzu, Kyoto, Japan), and Scanning Electron Microscope Inspect-S50 (FEI, Hillsboro, OR, USA).

### 2.2. Preparation of MIP

A total of 1 g of chitosan was added to 50 mL of 5% (*v*/*v*) acetic acid. The mixture was then stirred for 2 h at 50 °C, until a clear and homogeneous solution was obtained. Next, 1 mL of 0.1% (*w*/*v*) *p*-AP solution was added into 7 mL of chitosan solution, and the mixture was stirred and then 1 mL of 2% (*v*/*v*) glutaraldehyde or STPP 2% (*w*/*v*) was added. The mixture was stirred at room temperature for 1 h, followed by the addition of 1 mL polyvinyl alcohol 1% (*w*/*v*). After the mixture was stirred for a few minutes, it was ready to be used as an SPCE modifier.

### 2.3. MIP Characterization

The characterization was carried out on a MIP with a *p*-AP concentration of 1.0% (M_4), with crosslinkers, both glutaraldehyde (M-4G) and STTP (M-4S). MIP properties was characterized based on the FTIR spectra and MIP surface morphology by scanning electron microscopy (SEM). The MIP preparation procedure was identical to (2.2), but instead, after the mixture was heated at room temperature for 1 h, it was then poured into a Petri dish and heated at 50 °C to form a thin layer. Next, the thin layer was removed from the Petri dish and washed with 0.1 M NaOH followed by distilled water until the washing water was neutral, for the extraction of the template. The MIP was then dried until it was free of water. The FTIR analyses were carried out at the Chemistry Department of Brawijaya University, while the SEM analyses were carried out at the State University of Malang.

### 2.4. Modification of SPCE

A total of 20 µL of MIP (from Section 2.2) was dripped on the WE surface in SPCE, and then smoothed with a small brush. The coating was carried out twice, in which the first coating was heated at 50 °C for 2 min, whereas the second coating was heated at 50 °C for 5 min. A thin layer was formed on the WE surface, and then the surface was dripped with 50 µL of 0.1 M NaOH several times, followed by 50 µL of distilled water, until the water was not alkaline (which was checked by Litmus paper). This was carried out carefully so that the thin film was not damaged or loose.

### 2.5. p-AP Sensor Testing

Sensor evaluation was performed using a 50 µM *p*-AP solution which was dissolved in phosphate buffer pH 6.2. Evaluation of the effect of *p*-AP concentration on MIP was carried out by square wave voltammetry (SWV) at an amplitude potential of 50 mV, a frequency of 10 Hz, and a potential step of 10 mV, while the influence of the type of crosslinker was evaluated by cyclic voltammetry (CV) at −1.0 to 0.8 volts with a scan rate of 50 mV/s and potential step 10 mV. In both methods, 200 µL of the *p*-AP solution was dripped on the SPCE-modified surface.

### 2.6. p-AP Performance

Standard *p*-AP solutions in several concentrations from 0.5 to 50.0 µM were prepared in 0.1 M phosphate buffer pH 6, then determined by SWV using SPCE-M-4G, SPCE-M-4S, and SPCE. In addition, a sample solution from commercial paracetamol tablets was prepared. The tablets were weighed and then crushed. A total of (500.0 + 0.1) mg of the sample was dissolved in phosphate buffer pH 6, filtered off, and dissolved again in a 10 mL volumetric flask. The SWV was recorded at −0.15 to 0.20 Volts.

## 3. Result and Discussion

### 3.1. Preparation of MIP

In the synthesis of MIP, optimization of concentration of *p*-AP, as a template, and selection of cross-linking reagents, glutaraldehyde, and STPP, were studied. Glutaraldehyde as a cross-linker was used in the optimization of *p*-AP concentration. The concentrations of *p*-AP were 0; 0.1; 0.5; 1.0; 1.5; and 2.0% (*w*/*v*) in MIP. To simplify, the six MIPs were coded as M-1 to M-6. Evaluation results of the six MIPs, using square wave voltammetry (SWV), are presented in Figure 1 and Table 1.

From the voltammogram data (Table 1), it can be seen that there are two peaks for *p*-AP at SPCE without modification: 0.10 and 0.26 volts vs. Ag/AgCl. The peak potential (E_p_) of *p*-Ap, resulting from modified SPCE, is consistent at 0.09 volts. There are two stages of *p*-AP oxidation (Figure 2): the first was from *p*-AP to quinonimine and continued to quinone [41,42]. In the second stage of oxidation, a broad peak was not detected in the modified SPCE. Thus, the MIP membrane on the SPCE surface can cause the second oxidation undetected. This is due to the presence of a thin-film layer on the SPCE surface, which may inhibit the electron transfer from WE to CE at the oxidation of quinonimine to quinone; consequently, the peak current did not appear.

The peak current (I_p_) on the voltammogram is inversely related to the *p*-Ap concentration in the MIP, except at 1.0% (M-4). At 1.5 and 2.0% *p*-AP (M-5 and M-6), the voltammogram peaks were not clearly observed. SPCE-M_4 resulted in the highest peak current and the most symmetrical voltammogram shape. Based on the data in Table 1, the peak current from M-1 is higher than that of M-2 and M_3. In M-1, there is no *p*-AP, but there is still cross-linking of chitosan by glutaraldehyde, which forms a cavity; hence, *p*-AP can pass through to diffuse to the WE surface. In M-2 and M-3, it is possible that there is still *p*-AP remaining, which was not released during the washing, thus blocking the diffusion of *p*-AP from the bulk solution to WE during measurement. The *p*-AP in M-4 is 2 times higher than in M-3; thus, more templates are presumably formed, which then cause the peak current to be higher. However, if there is too much *p*-AP in the MIP, as in M-5 and M-6, the diffusion is more likely to be blocked.

Glutaraldehyde and STPP were compared as chitosan crosslinkers in MIP synthesis at 1% *p*-AP concentration. The two MIPs were used as SPCE modifiers and tested by cyclic voltammetry for 50 µM *p*-AP in pH 6.2 buffer solution. M-4G and M-4S are glutaraldehyde and STPP as crosslinkers, respectively, and the *p*-AP concentration in MIP is 1%. As shown in Figure 3, the oxidation–reduction reaction of *p*-AP is reversible in the SPCE-unmodified instance, with I_pa_/I_pc_ = 1 and ΔE_p_ = 60 mV. The I_pa_/I_pc_ for SPCE modified by M-4G and M-4S, are 1.6 and 1.2, respectively, whereas ΔE_p_ are 100 and 110 mV, respectively. Modification of SPCE by M-4G and M-4S causes *p*-AP to be oxidized more slowly; E_p_ shifted to the positive direction, and so ΔE_p_ increased and the oxidation–reduction reaction of *p*-AP showed a quasi-reversible property.

The identification of functional groups based on FTIR spectra, compared between chitosan, M-4G, and M-4S, is presented in Figure 4. There is no difference in functional groups between chitosan and M-4S. Meanwhile, in the FTIR spectrum for M-4G, there is a difference in the peaks at wave numbers of 1563 and 1402 cm^−1^, which indicates the presence of an -NH secondary amine group and tertiary alcohol [43]. Both M-4G and M-4S lost the peak at 575 cm^−1^ from chitosan, indicating an -OH out-of-plane bend [43], and that cross-linking possibly occurred in the -OH group of chitosan. Peak wavenumbers and determination of chitosan functional groups, M-4G, and M-4S are presented in Table 2. The cross-linking between chitosan and STPP, which occurs at pH 3, is an electrostatic interaction between the protonated amine group of chitosan (-NH_3_^+^) and the phosphate ion in STPP. The washing process by NaOH solution during the SPCE modification was predicted to cause the breakdown of the cross-links. This is indicated by the FTIR spectrum of M-4S which is identical to that of chitosan.

Surface morphology assessment of M-4G and M-4S by SEM, Figure 5) confirms that the *p*-AP is printed on the surface of M-4G, with no cavity formation observed, which is one of the characteristics of MIP formation. Meanwhile, on the surface of M-4S, the printed *p*-AP is not visible; only cavities are formed on the surface. This shows that the two crosslinkers have their own weaknesses. Chitosan cross-linking by STPP occurs due to electrostatic interactions and is highly dependent on pH, in which an increase in pH can cause a decrease in the number of cross-links. Chitosan cross-linking by glutaraldehyde occurs in the unprotonated chitosan amine group (-NH_2_); this reaction occurs more frequently at non-acidic pH, but at pH > 6.2, chitosan would have precipitated out. Therefore, for the synthesis of MIP using glutaraldehyde as a crosslinker, the pH of chitosan in acetic acid was adjusted to 5. Figure 6 shows an illustration of the formation of MIP using glutaraldehyde and STPP as the cross-linker.

In the MIP preparation, chitosan was hydrolyzed in acid (acetic acid) to produce chitosan with shorter chains, but not monomers. Therefore, in this process, chitosan was added with acetic acid and heated for about 2 h to produce a clear mixture. *p*-AP was added before glutaraldehyde to form a complex, or the interaction between chitosan and *p*-AP, the crosslinking reaction of chitosan by glutaraldehyde, was expected to be more directed so that the formation of MIP could occur properly. PVA was added last, as a film reinforcement, and was not involved in the reaction. The condition of chitosan cross-linking by glutaraldehyde was different from that of chitosan by STPP. Cross-linking of chitosan by glutaraldehyde occurs at a not too acidic pH, but at a pH ≤ pK_a_ of chitosan (6.2), so that the fraction of the -NH_2_ group in chitosan can be ≥50%. It was necessary to adjust the pH (≈5) before adding glutaraldehyde. The condition of cross-linking reaction of chitosan by STPP takes place at pH 3; thus, it is unnecessary to adjust the pH. STPP is alkaline; hence, when the STPP is added, it will change the pH of the mixture. The formation of MIP, both by glutaraldehyde and STPP, is illustrated in Figure 6. The illustration is modified from various literature. To release *p*-AP from MIP, template formation was carried out by washing MIP with 0.1 M NaOH solution, followed by distilled water to rinse off the remaining NaOH. This process was performed at MIP on a modified SPCE surface.

### 3.2. Performance of p-AP Sensor

Based on the results in the section above, SPCE-M-4G was chosen as the best sensor. The sensor was tested in 50 µM *p*-AP solution. Sensor performance can be maximized if the pH of the *p*-AP solution is suitable for oxidation, as shown in Figure 2, in which the oxidation of *p*-AP is affected by pH. The *p*-AP is a weak acid with pK_a1_ = 5.48 and pK_a2_ = 10.46, and can be oxidized under acidic conditions. In this study, optimization of pH in the range of 3–8 was carried out by a mixture of citrate and phosphate buffers. Figure 7 shows the voltammogram of a 50 µM *p*-AP solution at various pHs. As observed in Figure 6, peak current increases in direct proportion to pH in the range 3–6. The peak current decreased at pH 7 and 8. This can be explained by the illustration in Figure 8; at pH < 5.48, the -NH_2_ group on *p*-AP protonates to -NH_3_^+^, to form structure A. At pH > 5.48, the -NH_3_^+^ in *p*-AP decreases to form structure B. Oxidation of *p*-AP occurs in structure B, as shown in Figure 2, thus the peak current increases from pH 3 to 6. In theory (Figure 8D), from pH 7 to 9, the B structure of *p*-AP remains dominant (>95%), but *p*-AP oxidation occurred under acidic conditions; hence, the peak current of *p*-AP decreased at pH 7, whereas at pH 8, the peak current was not identified. The highest *p*-AP peak current was obtained at pH 6.

As presented in Figure 8D, the mole fraction of B at pH 3 is very small, only 0.3%, causing the slow diffusion rate of B and thus a higher potential is required for oxidation of B. At pH 4 to 5, B species increased, respectively, to 3.2 and 24.9%; thus, the rate of diffusion increases, and the potential required for oxidation decreases. Similarly, this also happened at pH 6 (76.8%) and 7, where species B was 76.8 and 97.0%, respectively. Therefore, E_p_ is inversely related to pH at 3 to 7, in which the linear regression of the relationship is: E_p_ = 0.119–0.011 pH with R^2^ is 0.995.

Quantitatively, the relationship between *p*-AP concentration and peak current (I_p_) is shown in Figure 9, in the range of 0 to 50 µM. Figure 9 shows that the linear concentration range of the SPCE-M-4G sensor is 0–35 µM, while for SPCE-M-4S and SPCE, it ranges from 0 to 50 µM. The sensitivity of SPCE-M-4G is the highest compared to that of SPCE-M-4S and SPCE (Table 3). SPCE-M-4G has the highest sensitivity compared to the others because M-4G contains *p*-AP, as shown in Figure 4a, which can be oxidized during measurement and then triggers the diffusion of *p*-AP from the bulk to the WE surface. Meanwhile, in SPCE-M-4S, the presence of M-4S can block diffusion of *p*-AP to the WE surface; hence, the sensitivity of SPCE-M-4S is lower than that of SPCE. Comprehensively, SPCE-M-4G performance was the best, with sensitivity of (3.7 ± 0.1) µA/µM, limit of detection of (2.1 ± 0.1) µM, and limit of quantification of (7.5 ± 0.1) µM, at a linear concentration range of 0 to 35 µM. The short linear concentration range is probably caused by the presence of *p*-AP in M-4G, which has not been released.

Accuracy was determined by standard addition to real samples. Out of the three samples tested, only one sample gave a positive signal, which was an expired drug sample, sample F. Figure 10 shows that sample F produced a signal, and the signal increased very sharply for sample F plus *p*-AP standard (F + 7µM), with recovery of *p*-AP concentration being (94.11 ± 0.01)%. It can be concluded that the *p*-AP sensor based on chitosan MIP can be applied to detect *p*-AP in paracetamol samples.

Based on the results of this study, to improve the performance of SPCE-M-4G, it is necessary to optimize PVA because it is likely that PVA can inhibit the release of *p*-AP from M-4G when washing with NaOH. The presence of *p*-AP on M-4G which has not been released is advantageous on one hand, but it has an impact on short linear concentration range. With the PVA optimization in the MIP preparation, it is hoped that the best composition will be produced to obtain the best sensor. In addition, in order to guarantee that MIP is not easily separated from the SPCE, it is necessary to develop a sensor manufacturing technique by adding MIP directly to the carbon ink for WE.

## 4. Conclusions

Electrochemical sensors can be developed from MIPs based on chitosan polymers, especially for *p*-aminophenol sensors. As a template of an MIP, the best *p*-aminophenol concentration was 4% in an MIP. Glutaraldehyde as cross-linker gave better sensitivity to the sensor than sodium tripolyphosphate. When identifying MIPs based on the FTIR spectrum, there was a change in the functional group in MIPs with glutaraldehyde as a crosslinker; in contrast, there was no change in the functional group when using STPP as a crosslinker. Based on identification of surface morphology from SEM images, *p*-aminophenol binds stronger to MIP with glutaraldehyde as a crosslinker than with STPP. The *p*-aminophenol sensor produced in this work has a working concentration range of 0.5–35 µM, a sensitivity of (3.7 ± 0.1) µM/µA, LoD of (2.1 ± 0.1) µM, LoQ of (7.5 ± 0.1) µM, and an accuracy of (94.11 ± 0.01)%. The sensors can be applied to paracetamol drug samples after several pre-treatments. From this research, development of a disposable p-AP sensor for the determination of *p*-AP in paracetamol drug samples remains possible.

## Figures and Tables

**Figure 1 polymers-15-01818-f001:**
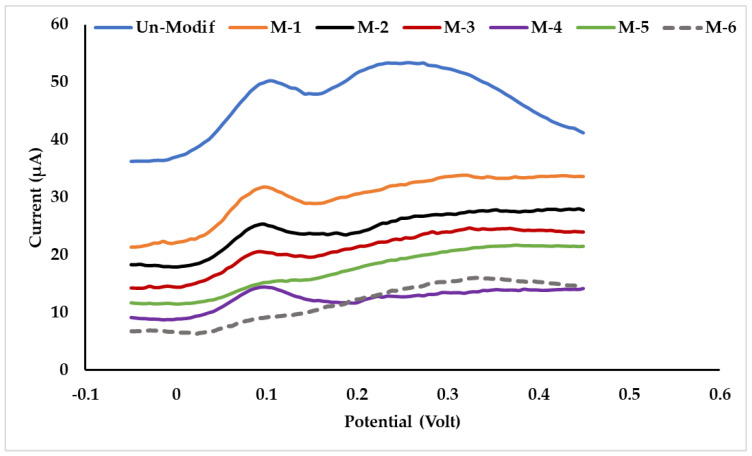
Voltammogram of *p*-AP 50 µM in buffer solution pH 6.2. Data were obtained from sensors for various *p*-AP concentrations in the MIP. SWV parameters at potential step 10 mV; frequency 10 Hz; and scan rate 50 mV/s.

**Figure 2 polymers-15-01818-f002:**
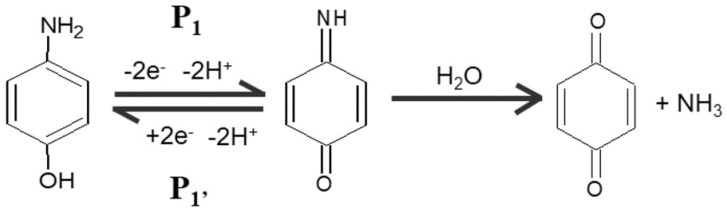
Oxidation of *p*-AP to quinonimine and quinone [42].

**Figure 3 polymers-15-01818-f003:**
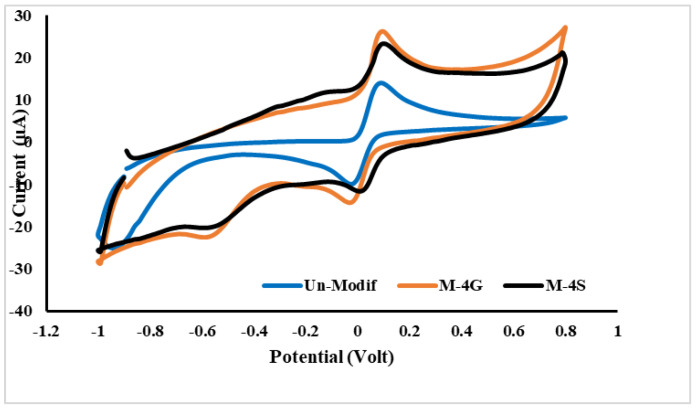
Voltammograms of cyclic voltammetry for *p*-AP solution in pH 6.2 buffer on SPCE-unmodified, SPCE-M-4G, and SPCE-M-4S.

**Figure 4 polymers-15-01818-f004:**
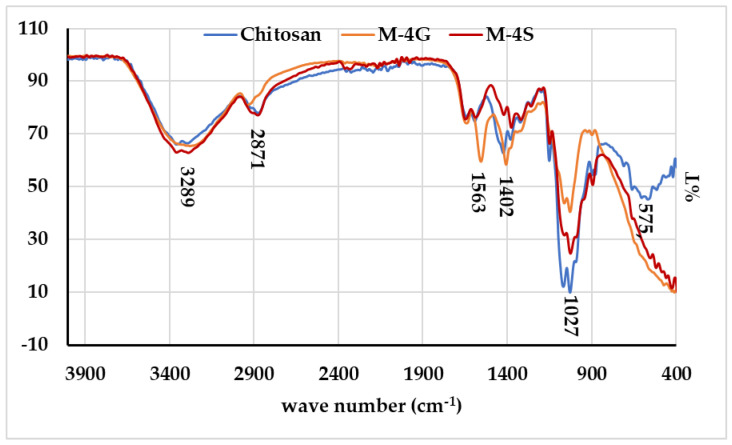
FTIR spectra for chitosan, MIP-glutaraldehyde (M-4G) and MIP-STPP (M-4S).

**Figure 5 polymers-15-01818-f005:**
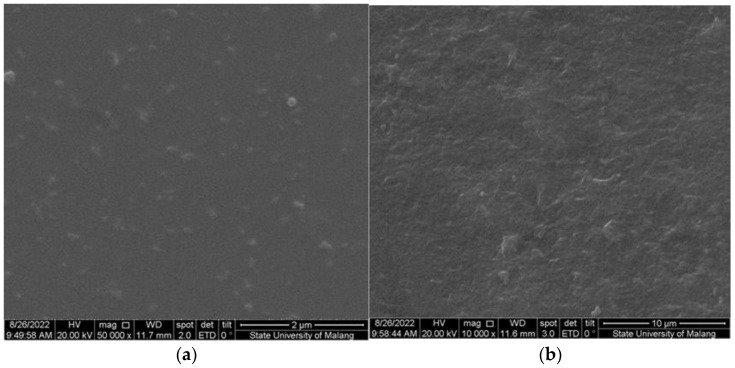
SEM image of the M-4G (**a**) and M-4S (**b**) surfaces.

**Figure 6 polymers-15-01818-f006:**
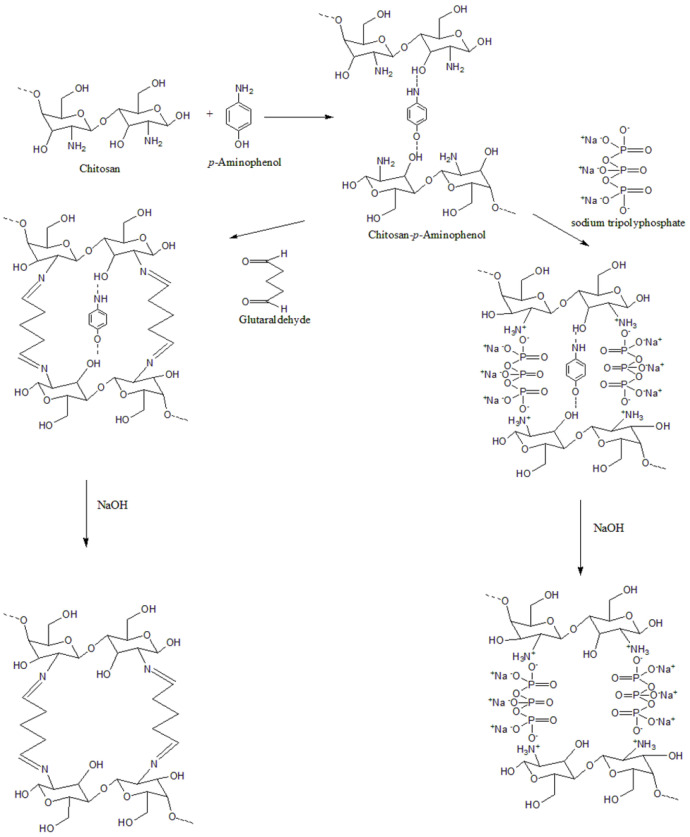
Illustration of the formation of chitosan-based MIPs by glutaraldehyde and STPP as crosslinkers [44].

**Figure 7 polymers-15-01818-f007:**
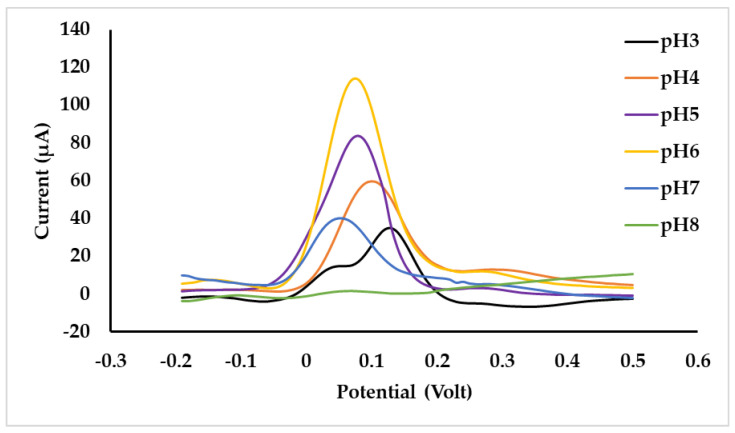
SWV voltammogram of *p*-AP 50 µM solution at various pHs on SPCE-M-4G.

**Figure 8 polymers-15-01818-f008:**
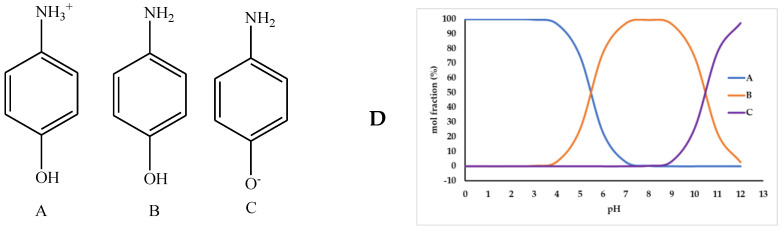
Structure of *p*-AP at pH < pK_a1_ (**A**); pK_a1_ < pH < pK_a2_ (**B**); pH > pK_a2_ (**C**) and mole fractions of A, B, and C (**D**).

**Figure 9 polymers-15-01818-f009:**
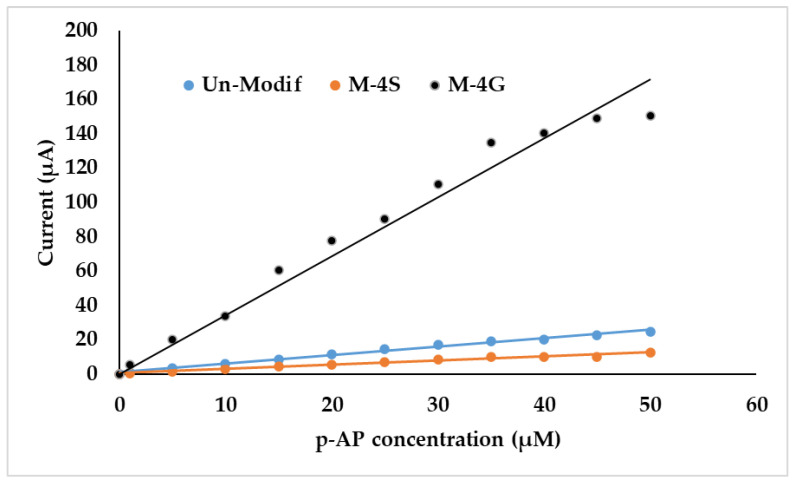
Linear regression of the relationship between *p*-AP concentration and anodic peak current. which is obtained from unmodified SPCE, SPCE-M-4G, and SPCE-M-4S.

**Figure 10 polymers-15-01818-f010:**
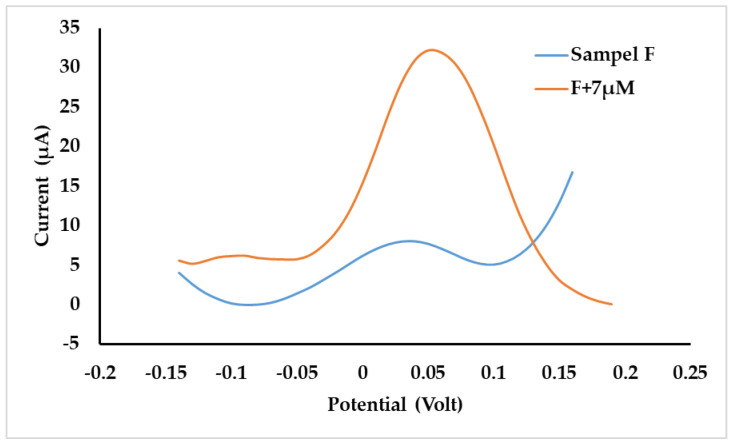
Voltammogram sample F and sample F + 7 µM standard of *p*-AP.

**Table 1 polymers-15-01818-t001:** Peak potential and peak current for 50 µM of *p*-AP solution resulted by several sensors in variation of template (*p*-AP) concentration.

Sensor	*p*-AP Concentration (%)	Peak Potential (Volt)	Peak Current (µA)
SPCE	-	0.101825	7.7
	-	0.262958	8.6
M-1	0.0	0.096790	9.4
M-2	0.1	0.091754	8.3
M-3	0.5	0.096790	7.9
M-4	1.0	0.091754	15.0
M-5	1.5	0.091754	1.7
M-6	2.0	0.091754	1.0

**Table 2 polymers-15-01818-t002:** Wavenumbers and functional groups of chitosan and MIPs.

Wavenumbers (cm^−1^)	Functional Groups	Chitosan	M-4G	M-4S
3289	O-H	+	+	+
2871	-N-CH_3_	+	–	+
1563	-NH secondary amine	–	+	–
1402	-OH tertiary alcohol	–	+	–
1027	C-N primary amine	+	+	+
575	-OH out of plane	+	–	–

**Table 3 polymers-15-01818-t003:** Sensor performance parameters from SPCE; SPCE-M-4S, and SPCE-M-4G.

Sensor	Linear Regression	R^2^	Sensitivity (µA/µM)	LoD (µM)	LoQ (µM)
SPCE	y = 0.49x + 1.07	0.9907	0.5 ± 0.1	3.1 ± 0.1	15.3 ± 0.1
SPCE-M-4S	y = 0.24x + 0.53	0.9779	0.2 ± 0.1	6.0 ± 0.1	25.0 ± 0.1
SPCE-M-4G	y = 3.74x + 0.68	0.9957	3.7 ± 0.1	2.1 ± 0.1	7.5 ± 0.1

## Data Availability

The data presented in this study are available in this same article.

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
