# Peer review of "Application of Chitosan-Based Molecularly Imprinted Polymer in Development of Electrochemical Sensor for p-Aminophenol Determination"

_polymers, 2023, doi:10.3390/polym15081818_

Round 1

Reviewer 1 Report

The manuscript “Application of Molecularly Imprinted Polymer Chitosan Based in Development of Electrochemical Sensor for p-Aminophenol Detection’’ is interesting work and well -written. But, it requires some minor corrections for the publication:

1.       Page 1, line 10: “Functional polymers that……………………. for acid sensors. veins [14]. What are veins?

2.       In manuscript, analyte was designated as para-aminophenol and sometimes 4-aminophenol. The name of the analyte should be uniform throughout the manuscript.

3.       Material and methods : the detail about electrochemical workstation has not been mentioned properly.

4.       Kindly correct grammatical issues.

Author Response

we've revised

Reviewer 2 Report

The article can be accepted after minor revision

1. The authors need to explain the advantages of carbon materials over other electrodes in the introduction part

2. The LOD should be written in the same units either micro or nanomolar in the comparison table

3.  If possible please include sensitivity table

4.  Please mention range of the concentration variation range

5. The conclusion should be further improved with important results

6.  What are the advantages of the article regarding sensor developments?

7. How did you derive the calibration curve (On the basis of peak current or peak height)?

8. What is the error limit in the calculation of LOD and LOQ?

9. Electrode sensitivity is missing?

10. Glassy carbon electrode is sensitive, why do you use this electrode?

11. Check the value of diffusion coefficient of K4[Fe(CN)6]

12. Diffusion/adsorption controlled process of the electrode should be further clarified by using the slope of log Ip vs. log v

13.  Is it Ret or Rct of the electrode?

Author Response

We have accommodated some of the suggestions, we have revised the manuscript

Reviewer 3 Report

In this research, an electrochemical sensor has been developed for the determination of p-aminophenol by modifying a screen-printed carbon electrode with chitosan-based molecularly imprinted polymers. The study is interesting, but the manuscript is very incomplete. The validation of the analytical method is very incomplete. New studies (interfering, stability, repeatability) need to be provided. Determination in samples is very confusing. The authors do not make it clear which samples were used. The experimental section of the manuscript is very poor in information. The novelty of the manuscript needs to be clarified. Therefore, I recommend that a major review be performed before the manuscript can be considered for publication in Polymers.

General comments:

1. The use of a chitosan-based MIP on SCPE for electroanalytical determination of butylated hydroxy anisole was published in Talanta, 223 (2021) 121689. Thus, a comment should be made about this work in the Introduction and reinforce the novelty of the present manuscript.

2. An important review work on chitosan-based MIPs for application in electrochemical sensors was published in TrAC Trends in Analytical Chemistry, 130 (2020) 115982. It remains a reading suggestion.

3. A study of interferents, repeatability, and stability must be provided in the manuscript.

4. The abbreviations chosen by the authors throughout the text (SPCE_M4G, M_4G,  M_4S, SPCE-M_4…) are very confusing. This must be reviewed.

5. Figure captions are very poor in information. Improve the captions of the figures.

6. The authors do not explain how MIPs increase current responses to p-AP.

7. How many measurements are possible to perform using an electrode? That is, without discarding the electrode and the need to form new films.

8. Update the terminologies for the electrochemical methods according to new IUPAC recommendations. See the information in Pure and Applied Chemistry, 92 (2020) 641–694. Current is I (in italics).

9. The authors, like many others, confuse the terms "detection" and "determination". Detection is qualitative by nature, while determination always is quantitative. Qualitative analysis is the detection of the presence of ions or compounds in an unknown sample, for example. The term "determination" refers to quantitative analysis to obtain data on the amount of analyte by weight or by concentration of an element or a compound in a sample. Therefore, most of the words “detection" in the manuscript should be replaced by the term "determination" (or "quantitation" or "assay") if quantitative assays are involved.

Specific comments:

1. Keywords:

a. Page 1. Keywords: glutaraldehyde; sodium tripolyphosphate; polyvinyl alcohol are not attractive. Use attractive keywords so that the article is easily found on search engines.

2. Introduction:

a. Page 2. Replace "Since p-AP is oxidative" with "Since p-AP is an electroactive compound".

b. Page 2. It is important to add the advantages of using electrochemical methods. So, add the sentence: “In that respect, electrochemical methods offer the benefits of elevated sensitivity, accuracy, convenient operation, cheap instrumentation, facile integration, and portability”. Add references Chemosensors, 10 (2022) 357 and Materials, 16 (2023) 1024 to validate this information.

c. Page 2. Do not use the term "calibration curve" if the graph is a straight line. Use the term "calibration plot".

d. Page 2. A buffer is a solution that can resist pH change upon the addition of acidic or basic components. So, the term "buffer solution" is redundant, just replace it with "buffer".

e. Page 2. The term "detection limit the minimum" is redundant. Just replace it with "detection limit".

f. Page 2. An important advantage of chitosan is the ability to form stable films on substrates (electrodes). Thus, add the sentence: "Furthermore, chitosan has the ability to form stable films on solid substrates." Add the reference Talanta, 252 (2023) 123836 to validate this information.

3. Experimental:

a. Pages 2 and 3. Standardize the description of the equipment: model (company, country).

b. Page 2. Inform the geometric area of the working electrode of the SPCE.

c. Page 2. Enter the molecular weight of chitosan (low, medium, or high). Also, inform the company and country.

d. Page 2. “…HCl (Sigma Alrdich).” Check the company name.

e. Describe how the MIP characterization experiments were carried out.

f. Inform the arrangement of electrodes available in the SPCE. What was the reference electrode available in the array?

g. You need to create a section to describe sample preparation. Describe the samples, and where they were acquired (company and city). How they were prepared for the analyses.

3. Results and discussion:

a. Page 3. “Two things that have been studied”. Things? Authors must use scientific language in the manuscript.

b. Page 3. “…resulted 2 oxidation peaks, at 0.10 and 0.26 volts.” Peak potentials are not absolute values. It is important to indicate which reference electrode was used. Use "at 0.10 and 0.26 V vs. Ag/AgCl (if this was the reference electrode).

c. Page 3. Indicate in the caption of Figure 1 the SWV parameters (amplitude, frequency and increment) used in the experiment.

d. Page 3. “There are two stages of p-AP oxidation, the first is p-AP to quinonimine and then to quinone, Figure 2, the second stage of oxidation produces a widened peak, and it cannot be detected in the modified SPCE. Thus, the presence of the MIP membrane on the SPCE surface, can result a selective oxidation reaction”. If bare SPCE can record both steps of the electrochemical reaction, and MIP/SPCE cannot, then MIP/SPCE is not selective. Authors need to proofread.

e. Page 3. “The peak current on the voltammogram is inversely related to the p-Ap concentration in the MIP, except for M_4”. Confusing statement. The authors need to improve the explanation.

f. Page 3. “SPCE-M_4 has produced the highest anodic current and the most symmetrical voltammogram shape, so that in subsequent studies the MIP composition according to M_4 has been used.” Why did the M_4 provide the highest current response? An in-depth discussion is needed.

g. Page 3. The x-axis of voltammograms needs to clarify which reference electrode is used. Example: Potential vs. Ag/AgCl / V.

h. Page 4. The potential values are with an absurd number of digits. Reduce significant figures to 3.

i. Page 4. “caused the oxidation reversibility of p-AP to decrease where Ipa/Ipc > 1”. Provide peak-to-peak potential separation data to discuss system reversibility. A system is considered reversible if the peak-to-peak potential separation equals 59.2 mV/z, where z is the number of electrons transferred.

j. Page 8. The recovery study shown by the authors is very confusing. Enhance Table 3. Add the p-AP concentration determined in the samples, the amount of p-AP added (spiked), the amount recovered, and the percentage recovered. Clearly describe which samples were used. Table 3 is very confusing.

Author Response

We have accommodated most of the suggestions, we have revised the manuscript

Reviewer 4 Report

The authors proposed a MIP-based screen-printed electrode for voltammetric detection of p-aminophenol. The MIP is prepared using chitosan as a functional monomer; glutaraldehyde and sodium tripolyphosphate as crosslinkers.

The paper, in my opinion, cannot be published in Polymers in the present form.

Some punctual observations on the critical issues of the manuscript are reported below.

- First of all, the English level is insufficient for publishing the manuscript in its present form. Some sentences and scientific terminology have to be improved.

- the procedure for preparing the prepolymeric mixture of the MIP has to be better detailed: "stirring for a while" and "heating for a while" cannot be helpful, and the authors have to explain in detail the method. The same goes for electrode functionalization.

- About the "optimization," as underlined in several reviews, books, and papers (see, for example, "S. A.  Weissman and  N.  G.  Anderson, Design of Experiments (DoE) and Process Optimization. A Review of Recent Publications, Org. Process Res. Dev. 2015, 19, 1605−1633), the traditional optimization approach, varying one factor/variable at a time (OFAT, also called OVAT), suffers from many drawbacks. On the other hand, statistical design of experiments (DoE) is a powerful approach to optimizing chemical processes and methods. The systematic approach inherent in DoE eliminates researcher bias and often will lead to experimental conditions that one had not considered previously. Moreover, a significant advantage of using DoE is the ability to quickly detect how interactions between factors can affect product yield and quality. Also, by simultaneously varying parameters, the DoE approach can be more efficient than the traditional method of changing one factor/variable at a time.

Author Response

We have tried to improve the English grammar

Round 2

Reviewer 3 Report

Doubts were resolved, and the manuscript was improved. Therefore, I recommend that the manuscript be accepted

Author Response

The response from the author is presented in a table, in a file

Reviewer 4 Report

The authors improved the manuscript, but the English level is still unsuitable for a scientific journal.

In my opinion, after a careful revision of the English language and style, the manuscript can be accepted.

Author Response

(The authors gave the same response as above.)
